# Can care staff accurately assess health-related quality of life of care home residents? A secondary analysis of data from the OPERA trial

Ben Parker, Stavros Petrou, Martin Underwood, Jason Madan

► Prepublication history and additional material is available. To view please visit the journal (http://dx.doi.org/10.1136/bmjopen-2016-012779).

Warwick Clinical Trials Unit, University of Warwick, Coventry, UK

**Correspondence to**
Ben Parker;
B.Parker@warwick.ac.uk

## ABSTRACT

**Objectives:** To compare assessments of health-related quality of life outcomes of care home residents reported by residents and care staff acting as proxies.

**Design:** Linear regression and bivariate modelling of paired assessments from care home residents and care staff.

**Setting:** 78 care homes in 2 regions in England.

**Participants:** 556 care home residents aged 65 years or older and care staff.

**Main outcome measures:** EQ-5D utility scores and responses to individual EQ-5D dimensions.

**Results:** The depression status, cognitive function, physical function, activities of daily living, social engagement, pain and dementia diagnosis of care home residents all predicted discrepancies in EQ-5D reporting. For residents with no depressive symptoms, care staff underestimated residents' mean EQ-5D utility score by 0.134 (95% CI 0.097 to 0.171) and for those with severe depressive symptoms they overstated mean utility scores by 0.222 (95% CI 0.104 to 0.339). With increasing levels of pain in residents the care staff progressively estimated EQ-5D utilities above self-reported values; by 0.236 (95% CI 0.003 to 0.469) in those with the second highest pain scores. For those with no cognitive impairment, proxies overstated mean utility scores by 0.097 (95% CI 0.049 to 0.146), while for those with severe cognitive impairment they underestimated mean utility scores by 0.192 (95% CI 0.143 to 0.241).

**Conclusions:** Care home residents and staff appear to differ fundamentally in their assessment of the health-related quality of life, as measured by the EQ-5D, of residents with different levels of depression, pain and/or cognitive impairment. This could lead to interventions evaluated using proxy-based quality-adjusted life year estimates being wrongly rejected on cost-effectiveness grounds and may also make it difficult for carers to act as advocates with health and social care professionals for certain groups of residents. A more resident-focussed approach to assessment of health-related quality of life is needed.

### Strengths and limitations of this study

- Care home staff commonly act as proxies for care home residents, providing health-related quality of life data on their behalf in research studies, and in clinical contexts.
- Data from a large clinical trial was used, with 556 paired assessments from care home residents and care staff.
- This study describes, for the first time, subgroups of care home residents for whom particularly large discrepancies exist between self-reported and proxy-reported utility scores.
- The results raise concerns around commissioning of services for these groups based on economic evaluations using proxy-reported utility scores.
- The study highlights the need for further research to identify if these discrepancies affect the ability of care home staff to act as the resident's advocate in encounters with health and social care services.

problematic. There is a high prevalence of dementia[1] and many residents are too frail or cognitively impaired to provide health-related quality of life data.[2] A 2014 review identified health-related quality of life instruments that have had their measurement properties validated for care home residents[3] recommending the use of the QUALIDEM for people with dementia[4–6] and the Psychosocial Quality of Life Domains questionnaire for those without dementia.[7] However, both the number and methodological quality of studies assessing the properties of health-related quality of life measures in this population are limited, and neither of these recommended measures are preference-based, constraining their use for economic evaluation purposes.[8]

Preference-based measures of health-related quality of life, such as the EuroQol EQ-5D-3L, can provide important information for

## INTRODUCTION

Measurement of the health-related quality of life of care home residents can be

resource allocation decisions. The EQ-5D is a generic preference-based measure of health-related quality of life and is recommended for use in cost-effectiveness studies by the National Institute for Health and Care Excellence (NICE) in England and Wales and the Washington panel on Cost Effectiveness in Health and Medicine in the USA.[9] [10] A potential solution to the problems associated with collecting health-related quality of life data in care home residents is to allow care home staff members to act as proxies and provide these data on behalf of residents. However, in relying on the perspective of care home staff to describe or value health states, discrepancies between the descriptions offered by care staff compared with those offered by residents themselves may lead to inconsistencies in resource allocation decisions.[11] The levels of inter-rater agreement between resident and proxy-reported EQ-5D scores in this setting has previously been examined, revealing poor to fair agreement at the level of the EQ-5D dimensions (cluster-adjusted κ −0.03 to 0.26), with moderate agreement at the utility score level (cluster-adjusted intraclass correlation coefficient of 0.44–0.50).[2]

This study extends this earlier research, using statistical models to explore which characteristics of care home residents predict discrepancies in reporting between residents and care home staff. These predictors are first investigated at the level of EQ-5D index (utility) scores, with a further dimension-level analysis exploring in which of the five EQ-5D dimensions these characteristics predict discrepancies. In doing so, this study adds depth to the current understanding around discrepancies between resident and proxy EQ-5D scores in care home settings, as well as insights into when and to what extent the health state descriptions offered by each deviate.

## METHODS

This study used data from the Older People's Exercise intervention in Residential and nursing Accommodation (OPERA) study, a cluster-randomised controlled trial with homes as the unit of randomisation. The OPERA study is described in detail elsewhere.[12–14] OPERA compared a whole home intervention targeted at increasing physical activity combined with a twice-weekly physiotherapist-lead exercise programme against a depression awareness programme for care home staff. The primary outcome measure was depression score as measured by the Geriatric Depression Scale-15 (GDS-15).

This analysis uses baseline data collected in the OPERA study. Prior to randomisation, study research nurses/physiotherapists administered a number of survey instruments to the care home residents participating in the trial, including the EuroQol EQ-5D-3L,[15] GDS-15,[16] Mini-Mental State Examination (MMSE)[17] and a measure of current pain (measured on a five-point ordinal scale).[12] Data on whether or not a diagnosis of dementia had been made was collected from medical records. There was also a brief physical assessment using the Short Physical Performance Battery

(SPPB).[18] [19] The resident's key member of staff, or carer looking after them on the day of data collection, was also asked to complete proxy versions of the EQ-5D-3L,[15] Barthel Index[20] and Social Engagement Scale (SES)[21] [22] on behalf of the resident. Typically carers in UK care homes do not have qualifications. Depending on the home the proxy EQ-5D responses were provided by a carer familiar with the resident or the care home manager. A summary of the key properties of these measures and their application in the OPERA study is provided in table 1.

The data set used for this study included 556 residents, comprising those individuals for whom both self-reported and proxy EQ-5D scores and dimension-level responses, as well as data for the covariates used in the statistical models, were available at baseline. A total of 504 participants in the OPERA study were excluded due to missing data.

Linear regression models were estimated using ordinary least squares, with the EQ-5D utility score regressed on scores for the GDS-15, MMSE, dementia diagnosis, Pain score, SPPB, Barthel Index and SES, with age, sex and length of stay included as controls. Age and sex are standard controls and have been used in health utility studies in a similar population,[23] while length of stay was included to control for the fact that those residents with a longer length of stay would likely have been known to their proxy for longer and may also have experienced some adaptation to the care home environment.

In seeking to understand differences between utility scores estimated from resident-reported and proxy-reported EQ-5D responses, the first point of interest was the extent of variation in the direction and magnitude of the coefficients on the prespecified covariates between the models predicting self-reported and proxy-reported utility scores. A further model used the same regression function for the difference between self-reported and proxy-reported utility scores. The strategy of this analysis was therefore to construct the conditional mean EQ-5D utility score in this manner, for residents and their proxies, and then to explore differences between the coefficients in each regression function. By then examining the statistical significance of these coefficients in the model predicting the difference between self-reported and proxy-reported utility score, the extent to which differences in each covariate explained deviations between utility scores produced by residents and their proxies could be assessed. The dimension-level analysis then proceeded to explore the drivers of these effects, at the level of the five individual dimensions of the EQ-5D (mobility, self-care, usual activities, pain and anxiety/depression). Owing to the potential effects of clustering by care home, all models specified cluster-robust SEs with the home specified as the clustering variable. This produces a variance–covariance matrix that is robust to clustering and also relaxes the assumption of homoscedasticity.[24–27]

**Table 1**  Summary of outcome measures in OPERA study

| Measure | Data source | Measure of | Higher score indicates/ score interpretation | Measurement details |
|---|---|---|---|---|
| EQ-5D-3L[15] | Resident/ proxy | Preference-based generic quality of life measure Responses are given in each of five 'dimensions' relating to health-related quality of life: mobility, self-care, usual activities, pain and anxiety or depression. Responses are one of three levels in each dimension; no problems, some or moderate problems, serious or extreme problems. A further question measures perceived current health state on a visual analogue scale. | Higher is better Higher score indicates higher 'utility'—see measurement details. | The dimension-level responses are mapped to a 'utility' score ranging from −0.59 (worse than death) to 1 (full health). The mapping algorithm is based on the results of a preference study conducted on a representative sample of the UK population.[38] |
| Depression (GDS-15)[16] | Resident | Measure of depressive symptoms designed for use in elderly populations | Higher is worse Higher score indicates greater number of depressive symptoms (ie, more depressed). *Interpretation*[34] 0–4: not typically cause for concern 5–8: mild depression 9–11: moderate depression 12–15: severe depression | Consists of 15 yes/no questions, producing a score ranging from 0 to 15 |
| Cognitive functioning (MMSE)[17] | Resident | Measure of cognitive function | Higher is better Higher score indicates higher level of cognitive functioning. *Interpretation*[33] 0–17: severe cognitive impairment 18–23: mild cognitive impairment 24–30: no cognitive impairment | Consists of 11 questions covering orientation, registration, attention and calculation, recall, and language and produces a score from 0 to 30 |
| Pain score[12] | Resident | Measure of current pain | Higher is worse Higher score indicates a higher level of current pain. | Presence or absence of pain was ascertained from the EQ-5D pain question. For those with pain their current level of pain was assessed on a |

Continued

**Table 1** Continued

| Measure | Data source | Measure of | Higher score indicates/ score interpretation | Measurement details |
|---|---|---|---|---|
| | | | | five-point ordinal scale ranging from no pain to pain as bad as it could be. |
| Dementia | Medical records | Diagnosis (or not) of dementia recorded in care home records | 0: no diagnosis<br>1: diagnosis | Data on whether or not the resident had been diagnosed with dementia were collected from medical records. |
| Physical functioning (SPPB)[18 19] | Resident | Measure of physical function | Higher is better<br>Higher score indicates greater physical functioning.<br>*Interpretation*[32] 0–3: severe limitations<br>4–6: moderate limitations<br>7–9: mild limitations<br>10–12: minimal limitations | Combines the results of the gait speed, chair stand and balance tests[19] and produces a score ranging from 0 to 10 |
| Activities of daily living (Barthel Index)[20] | Proxy | Measure of activities of daily living | Higher is better<br>Higher score indicates greater functional independence.<br>*Interpretation*[31] 0–20: total dependence<br>21–60: severe dependence<br>61–90: moderate dependence<br>91–99: slight dependence<br>100: independent | Consists of 10 items relating to daily functioning, producing a score ranging from 0 to 100 |
| Social engagement (SES)[21 22] | Proxy | Measure of social engagement | Higher is better<br>Higher score indicates greater social engagement. | Consists of six yes or no responses, generating a score from 0 to 6 |

GDS, Geriatric Depression Scale; MMSE, Mini-Mental State Examination; SES, Social Engagement Scale; SPPB, Short Physical Performance Battery.

A further analysis regressed EQ-5D responses at the dimension level on the same clinical variables and covariates detailed previously. Bivariate probit models were estimated, allowing for simultaneous estimation of the function predicting the resident and proxy response in each EQ-5D dimension. Marginal effects for each of the covariates in each function were computed, giving an estimate for the effect of a unit increase in a given covariate on the probability of the outcome. The dichotomous outcome in this case was a level 2 (some or moderate problems) or a level 3 (severe or extreme problems) response versus a level 1 (no problems) response in a given dimension of the EQ-5D (separate models were estimated for each of the five EQ-5D dimensions). The simultaneous estimation of the functions for residents and for proxies allowed the difference between these marginal effects to be tested for statistical significance, thus allowing the identification of those variables predicting a difference in resident and proxy response in each dimension. Cluster-robust SEs were used in these models as for the earlier models.

All statistical models were estimated in Stata SE V.14 (Statacorp. Stata Statistical Software: Release 14: College Station, TX: Statacorp LP, 2015) with plots produced in RStudio[28] using the R package ggplot2.[29]

## RESULTS

The study population was comprised of 556 care home residents, with an average age of 86 years. The average EQ-5D utility score using resident responses was 0.56, compared with 0.51 when using proxy responses. Using the diagnostic of a GDS-15 score of ≥5, indicating the presence of depressive mood,[30] 45% of study participants were depressed. 49% of study participants had a Barthel Index score indicative of 'severe dependence' or 'total dependence'.[31] 79% of study participants had an SPPB score indicating 'severe limitations'.[32] 40% of study participants had an MMSE score indicating 'severe cognitive impairment' (71% had a score indicating either mild or severe cognitive impairment)[33] and 23% of study participants had a diagnosis of dementia recorded in their care home records (table 2).

### Utility score analysis

By dividing the clinical variables considered into groups by clinical relevance, variation between resident and proxy-reported health-related quality of life outcomes between these groups was explored. Plots of three of these analyses are presented in figures 1–3, with the 95% SE bars showing the confidence in the mean estimates in each group. Also shown is the difference between the two means, with 95% SE bars from a paired t-test. The GDS-15 score has been split into groups corresponding to no (0–4), mild (5–8), moderate (9–11) and severe (12–15) depressive symptoms.[34] The MMSE score has been split into groups corresponding to severe (0–17), mild (18–23) and no (24–30) cognitive

**Table 2** Summary statistics for key variables

| | Mean (SD) | Range |
|---|---|---|
| Resident-reported utility | 0.56 (0.38) | −0.59 to 1 |
| Proxy-reported utility | 0.51 (0.32) | −0.35 to 1 |
| Depression (GDS-15) | 4.58 (3.14) | 0 to 14 |
| Cognitive functioning (MMSE) | 19.06 (6.42) | 2 to 30 |
| Activities of daily living (Barthel Index) | 60.67 (26.08) | 0 to 100 |
| Physical functioning (SPPB) | 2.07 (2.2) | 0 to 10 |
| Pain score | 1.64 (0.94) | 1 to 5 |
| Social engagement (SES) | 4.81 (1.64) | 0 to 6 |
| Age | 86.32 (7.45) | 65 to 107 |
| Length of stay (years) | 2.38 (2.6) | 0 to 27.8 |

GDS, Geriatric Depression Scale; MMSE, Mini-Mental State Examination; SES, Social Engagement Scale; SPPB, Short Physical Performance Battery.

impairment.[33] For those with pain, their current level of pain was assessed on a five-point ordinal scale ranging from no pain to pain as bad as it could be.[12]

Different levels of GDS-15 (depressive symptoms), MMSE (cognitive functioning), SPPB (physical functioning), Barthel Index (activities of daily living), SES (social engagement), Pain score and a diagnosis of dementia in care home records all showed a statistically significant association with differences between resident and proxy-reported EQ-5D utility scores (table 2). Typically, residents gave a more optimistic assessment of their health-related utility than their proxy counterparts.

Table 3 shows the results from the regression functions estimated to predict self-reported utility, proxy-reported utility and the difference between the two utility scores. Coefficients are displayed, with SEs in parentheses, and those variables found to be statistically significant predictors are starred. For all outcome measures, except GDS-15 (depressive symptoms) and the Pain score, a higher score indicates an improvement in health.

The statistically significant coefficients in the self-reported and proxy-reported regression functions were generally in the expected direction, with an improvement in a given measure predicting an increase in utility score. The exceptions to this were the MMSE and a diagnosis of dementia in care home records, with a single unit improvement in MMSE score predicting a 0.009 unit decrease in self-reported utility score, and a diagnosis of dementia predicting an increase in self-reported and proxy-reported utility scores (0.128 and 0.055, respectively).

Differences in GDS-15, MMSE, SPPB, Pain score and a diagnosis of dementia were all predictive of a greater change in self-reported rather than proxy-reported utility score. For example, a unit increase (worsening) in the GDS-15 was predictive of a 0.033 unit decrease in self-reported utility score, but only a 0.010 unit decrease in proxy-reported utility score. Differences in the Barthel Index, however, were predictive of a greater change in proxy-reported than self-reported utility score,

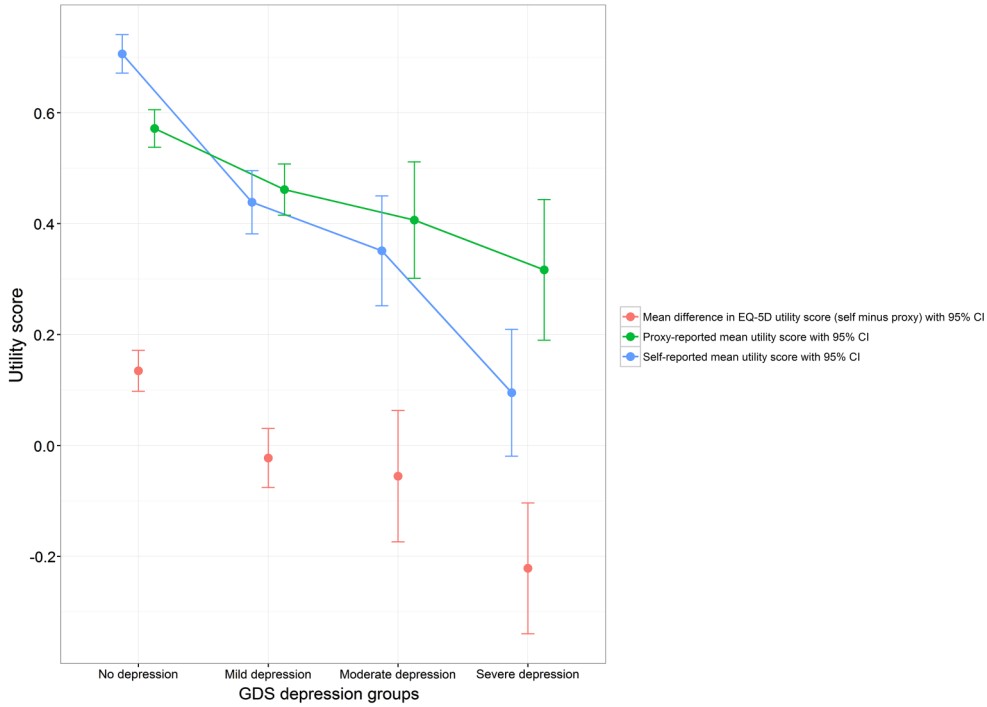

**Figure 1** **Self-report and proxy EQ-5D utility score for different levels of depressive symptoms (GDS).*,†**
*Regression coefficients for GDS (higher = more depressive symptoms) on EQ-5D utility score (statistically significant coefficients starred): −0.0328* (self-report), −0.0101* (proxy), −0.0227* (self minus proxy)
†Sample size in each group: 305 (no depression), 181 (mild depression), 48 (moderate depression), 22 (severe depression).

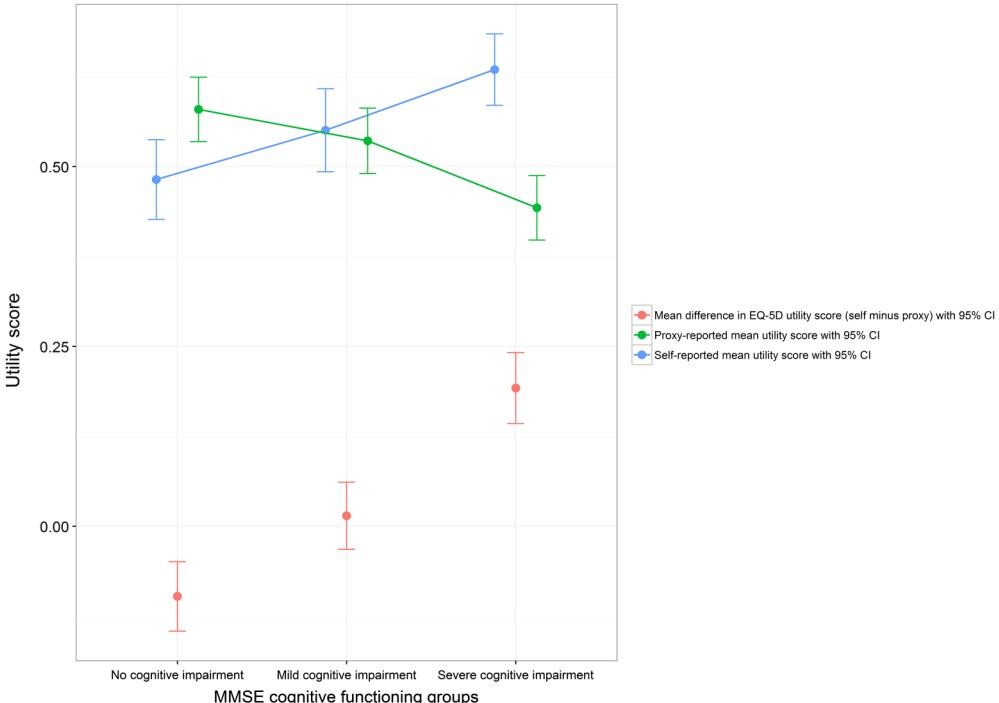

**Figure 2** **Self-report and proxy EQ-5D utility score for different levels of cognitive impairment (MMSE).*,†**
*Regression coefficients for MMSE (higher = better cognitive functioning) on EQ-5D utility score (statistically significant coefficients starred): −0.0089* (self-report), 0.0007 (proxy), −0.0096* (self minus proxy).
†Sample size in each group: 162 (no cognitive impairment), 172 (mild cognitive impairment), 222 (severe cognitive impairment).

**Figure 3** Self-report and proxy EQ-5D utility score for different levels of Pain score (1 = no pain, 5 = pain as bad as it could be).*,†

*Regression coefficients for Pain score (higher = worse pain) on EQ-5D utility score (statistically significant coefficients starred): −0.1500* (self-report), −0.0502* (proxy), −0.0997* (self minus proxy).

†Sample size in each group: 352 (Pain score = 1), 73 (Pain score = 2), 112 (Pain score = 3), 15 (Pain score = 4), 4 (Pain score = 5).

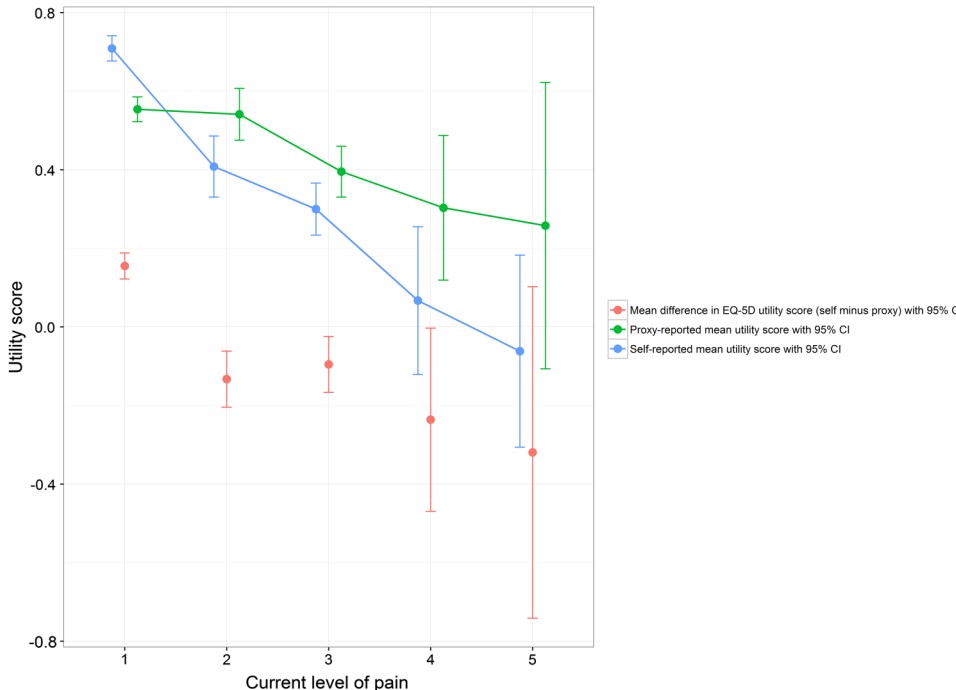

Mean difference in EQ-5D utility score (self minus proxy) with 95% CI
Proxy-reported mean utility score with 95% CI
Self-reported mean utility score with 95% CI

**Table 3** Model coefficients for self-reported and proxy-reported EQ-5D utilities

| Variable | Self-reported | Proxy-reported | Difference (resident-proxy) |
|---|---|---|---|
| Depression (GDS-15) | −0.0328† (0.0035) | −0.0101† (0.0035) | −0.0227† (0.0048) |
| Cognitive functioning (MMSE) | −0.0089† (0.0018) | 0.0007 (0.0017) | −0.0096† (0.0019) |
| Physical functioning (SPPB) | 0.0203† (0.0065) | 0.0065 (0.0059) | 0.0138 (0.0071) |
| Activities of daily living (Barthel Index) | 0.0049† (0.0007) | 0.0078† (0.0005) | −0.0030† (0.0008) |
| Social engagement (SES) | −0.0103 (0.0064) | 0.0077 (0.0068) | −0.0180† (0.0087) |
| Pain score | −0.1500† (0.0127) | −0.0502† (0.0125) | −0.0997† (0.0160) |
| Dementia | 0.1278† (0.0265) | 0.0548† (0.0239) | 0.0730† (0.0327) |
| Sex | −0.0417 (0.0245) | 0.0159 (0.0208) | −0.0576 (0.0305) |
| Age | 0.0029 (0.0018) | 0.0020 (0.0015) | 0.0010 (0.0023) |
| Length of stay | 0.0058 (0.0039) | 0.0026 (0.0033) | 0.0032 (0.0051) |
| Constant term | 0.4691† (0.1890) | −0.1619 (0.1501) | 0.6310† (0.2217) |

†Statistically significant at the 5% level.
Sex: 0=female, 1=male.
Dementia: 0=no diagnosis, 1=diagnosis.
GDS, Geriatric Depression Scale; MMSE, Mini-Mental State Examination; SES, Social Engagement Scale; SPPB, Short Physical Performance Battery.

with a unit improvement predicting a 0.005 increase in self-reported utility score compared with a 0.008 increase in proxy-reported utility score. For two of the covariates (MMSE and SPPB), differences were associated with changes in the self-reported utility score only, and were not statistically significant in the regression function for the proxy-reported utility score. SES did not have a statistically significant association with differences in either self-reported or proxy-reported utility individually, but was associated with a difference between the two (a unit increase in SES was associated with proxy-reported utility being 0.018 higher on the EQ-5D utility cardinal scale than self-reported utility).

## EQ-5D dimension score analysis

Having explored clinical and sociodemographic predictors of variation between self-reported and proxy-reported EQ-5D utility scores, the analysis proceeded to explore how these predictors manifested at the level of the EQ-5D dimensions. For example, for those variables predicting a difference between self-reported and proxy-reported utility, does this impact manifest in a broadly uniform manner across all five dimensions or is it instead focussed on particular dimensions?

Table 4 shows the proportions of level 1 (no problems), level 2 (some or moderate problems) and level

**Table 4** Proportions of level 1, 2 or 3 responses in each dimension of the EQ-5D

| | Self-reported | | | Proxy-reported | | |
| --- | --- | --- | --- | --- | --- | --- |
| | No problems | Some or moderate problems | Severe or extreme problems | No problems | Some or moderate problems | Severe or extreme problems |
| Mobility | 0.33 | 0.53 | 0.13 | 0.34 | 0.54 | 0.12 |
| Self-care | 0.53 | 0.35 | 0.12 | 0.28 | 0.50 | 0.21 |
| Usual activities | 0.55 | 0.31 | 0.14 | 0.43 | 0.46 | 0.12 |
| Pain | 0.49 | 0.42 | 0.09 | 0.39 | 0.57 | 0.04 |
| Anxiety/depression | 0.60 | 0.35 | 0.05 | 0.52 | 0.43 | 0.04 |

3 (severe or extreme problems) responses in each EQ-5D dimension, using both resident and proxy responses.

Bivariate probit models were estimated for responses to each of the five dimensions of the EQ-5D, allowing simultaneous estimation of the regression functions for resident and proxy responses for each EQ-5D dimension. The primary results of this dimension-level analysis are shown in table 5. The table presents the marginal effects for those variables for which the marginal effect was statistically significant for either self-reported or proxy-reported scores or the difference between the two marginal effects was statistically significant. The difference between the marginal effects for self-reported and proxy-reported responses for each variable was tested for statistical significance and the resulting p values are displayed.

The dimension-level analysis showed that, when considering whether differences in each of the covariates had a statistically significant association with differences in dimension response in each of the EQ-5D dimensions, there was significant variation between resident and proxy responses. For example, increases in GDS-15 and Pain score (indicating more severe depression and pain, respectively) had a statistically significant association with an increased likelihood of residents providing a level 2 or 3 response in every dimension of the EQ-5D, whereas for proxies this association was only statistically significant in the anxiety or depression and pain dimensions, respectively. Increases in the Barthel Index (indicating greater functional independence) had a statistically significant association with a decreased likelihood of proxies providing a level 2 or 3 response in every dimension of the EQ-5D, whereas for residents this effect was only statistically significant in the 'physical' EQ-5D dimensions of mobility, self-care and usual activities. While changes in the other covariates were associated with impacts in different EQ-5D dimensions for residents compared with their proxies, the pattern was less clear.

In the analysis at the level of utility scores, six variables were predictive of discrepancies between self-reported and proxy-reported utilities. These variables were GDS-15, Barthel Index, MMSE, Pain score, SES and a diagnosis of dementia in care home records. At the

dimension level, one point of interest is in which dimensions of the EQ-5D do differences in these variables predict a discrepancy between self-reported and proxy-reported dimension response (with dimension response represented here as the percentage change in the probability of providing a level 2 or 3 response, given a unit change in a variable)? The answer varied depending on the predictor, with differences in GDS-15 having a statistically significant association with a discrepancy between self-reported and proxy-reported dimension response in every EQ-5D dimension except pain. Differences in Pain score were associated with a discrepancy in self-care, usual activities and pain dimensions, while differences in the Barthel Index were associated with discrepancies in all EQ-5D dimensions except anxiety or depression. For differences in MMSE, dementia and SES, discrepancies between residents and their proxies were restricted to one or two dimensions only.

These discrepancies were most frequently due to residents being more responsive with respect to differences in a given covariate (ie, the probability of them providing a level 2 or 3 response being impacted to a greater extent) than their proxies. The exceptions to this were given differences in the Barthel Index and the SES, where proxies exhibited the greater 'sensitivity'.

## DISCUSSION

The context of this study is that care home residents and staff, when asked to provide responses relating to residents' health-related quality of life using the EQ-5D instrument, often produce discordant responses. While discrepancies may appear small when considering a population as a whole (mean self-reported utility score of 0.56 compared with a mean proxy-reported utility score of 0.51 in our study), we show that much larger discrepancies exist in subgroups defined by variables reporting depressive symptoms, pain, activities of daily living, cognitive functioning, social engagement and a diagnosis of dementia in care home records. In the case of depression, pain and activities of daily living, these discrepancies are driven by differences that span multiple dimensions of health-related quality of life. Discrepancies in health state descriptions reported by

**Table 5** Marginal effects in each dimension of the EQ-5D, for residents and proxies

| | Marginal effect: self-reported* | Marginal effect: proxy* | p Value for difference between the two effects |
|---|---|---|---|
| **Depression (GDS-15)** | | | |
| Mobility | 0.0320† (0.0059) | 0.0055 (0.0050) | 0.0004† |
| Self-care | 0.0315† (0.0065) | −0.0036 (0.0053) | 0.0000† |
| Usual activities | 0.0425† (0.0068) | −0.0056 (0.0048) | 0.0000† |
| Pain | 0.0097† (0.0047) | −0.0052 (0.0081) | 0.0696 |
| Anxiety/depression | 0.0570† (0.0048) | 0.0209† (0.0076) | 0.0000† |
| **Pain score** | | | |
| Mobility | 0.0618† (0.0214) | 0.0134 (0.0201) | 0.0856 |
| Self-care | 0.0753† (0.0208) | 0.0065 (0.0155) | 0.0024† |
| Usual activities | 0.0683† (0.0192) | −0.0155 (0.0193) | 0.0011† |
| Pain | 1.3223† (0.0746) | 0.1559† (0.0201) | 0.0000† |
| Anxiety/depression | 0.0829† (0.0187) | 0.0223 (0.0258) | 0.0570 |
| **Activities of daily living (Barthel Index)** | | | |
| Mobility | −0.0025† (0.0010) | −0.0068† (0.0009) | 0.0005† |
| Self-care | −0.0051† (0.0009) | −0.0099† (0.0010) | 0.0005† |
| Usual activities | −0.0040† (0.0011) | −0.0074† (0.0009) | 0.0054† |
| Pain | 0.0001 (0.0007) | −0.0035† (0.0010) | 0.0019† |
| Anxiety/depression | −0.0010 (0.0010) | −0.0025† (0.0012) | 0.2657 |
| **Physical functioning (SPPB)** | | | |
| Mobility | −0.0426† (0.0074) | −0.0475† (0.0101) | 0.7057 |
| Self-care | −0.0443† (0.0107) | −0.0033 (0.0081) | 0.0001† |
| Usual activities | −0.0031 (0.0119) | −0.0223† (0.0099) | 0.1527 |
| Pain | −0.0174† (0.0071) | −0.0155 (0.0087) | 0.8641 |
| Anxiety/depression | 0.0053 (0.0117) | 0.0254† (0.0124) | 0.1482 |
| **Cognitive functioning (MMSE)** | | | |
| Mobility | 0.0182† (0.0033) | 0.0106† (0.0034) | 0.1008 |
| Self-care | 0.0157† (0.0029) | −0.0050 (0.0029) | 0.0000† |
| Usual activities | 0.0088† (0.0036) | −0.0042 (0.0031) | 0.0042† |
| Pain | 0.0011 (0.0029) | 0.0077† (0.0038) | 0.1241 |
| **Dementia** | | | |
| Mobility | −0.0707 (0.0444) | −0.1138† (0.0455) | 0.4523 |
| Self-care | −0.1310† (0.0470) | −0.0138 (0.0448) | 0.0586 |
| Usual activities | −0.1578† (0.0554) | −0.0561 (0.0503) | 0.1418 |
| **Social engagement (SES)** | | | |
| Usual activities | −0.0020 (0.0130) | −0.0508† (0.0102) | 0.0011† |
| Pain | 0.0256† (0.0108) | 0.0190 (0.0126) | 0.6631 |
| Anxiety/depression | 0.0029 (0.0117) | −0.0327† (0.0115) | 0.0095† |
| **Sex** | | | |
| Self-care | 0.0745† (0.0374) | −0.0203 (0.0392) | 0.0712 |
| Anxiety/depression | −0.0864† (0.0391) | −0.1429† (0.0488) | 0.3393 |
| **Age** | | | |
| Pain | −0.0018 (0.0019) | 0.0052† (0.0025) | 0.0137† |
| **Length of stay** | | | |
| Anxiety/depression | −0.0014 (0.0071) | −0.0186† (0.0074) | 0.0578 |

*Interpreted as the change in the probability (ie, 0.05 indicates a 5% increase in probability) of providing a level 2 or 3 (rather than a level 1) response in the given dimension, given a unit change in the variable.
†Statistically significant at 5% level.
GDS, Geriatric Depression Scale; MMSE, Mini-Mental State Examination; SES, Social Engagement Scale; SPPB, Short Physical Performance Battery.

residents and care staff were also present given differences in cognitive functioning, social engagement, physical functioning and age, although these discrepancies were present in fewer dimensions of health-related quality of life.

Depressed residents or those in pain were increasingly (with severity of depression or pain) more likely to report problems (as opposed to no problems) in every dimension of their health-related quality of life, whereas, when describing the health status of residents, carers were more likely to report residents as having problems only in the anxiety or depression, and pain dimensions, respectively. This suggests that carers, when assessing the health-related quality of life of care home residents, may

be unable to accurately replicate descriptions of health status provided by residents with depression and/or pain. While we have examined this within the context of a randomised controlled trial these findings are of major importance to how care is delivered to care home residents. Typically health and social care professionals rely on carers to be the advocates for care home residents. Our data indicate that when compared with self-report, care home staff are not able to accurately assess aspects of residents' health status as measured by the EQ-5D.

Residents with difficulties relating to their activities of daily living were more likely to report problems in the mobility, self-care and usual activities dimensions of the EQ-5D. Carers, however, were more likely than residents themselves to report residents as having problems in these dimensions, as well as being more likely to report residents as having problems in the pain and anxiety or depression dimensions, that is, dimensions in which residents own responses are unaffected by differences in self-care. Given the study setting (care homes), the ability of residents to care for themselves is clearly an aspect of the residents' health-related quality of life which is highly visible to carers. However, the results suggest carers place excess weight on the importance of this aspect to residents. In contrast, residents with improved physical function were less likely to report problems in the self-care dimension, whereas carers were no more or less likely to report problems in this dimension for this group of residents, suggesting care staff may not detect the importance of physical function in assessing resident's perceived ability to care for themselves. Increased social engagement was associated with carers being less likely to report that a resident had problems with their usual activities or with anxiety or depression, whereas level of social engagement had no impact on the residents' reports across these dimensions. However, the measure of social engagement was reported by care staff on the resident's behalf and so may be acting as a conduit for other aspects of residents' health-related quality of life, as perceived by care staff. Collectively, these results suggest that carers have a bias towards basing health-related quality of life assessments on those aspects of health-related quality of life with which they themselves are most likely to be involved (activities of daily living) as well as those aspects which may be most easily observed (carer-assessed social engagement), and that carers may mistakenly attribute changes across all dimensions of a resident's health-related quality of life in response to differences only in a resident's ability to carry out activities of daily living.

It would be expected that a worsening experience as measured by clinical measures would lead to a decrease in health utility and this was largely the case, with the two notable exceptions of cognitive functioning and a diagnosis of dementia in care home records. Somewhat counterintuitively, improvements in cognitive functioning were associated with a decreased self-reported health utility score, while a diagnosis of dementia was associated with an increased health utility score whether produced by residents or their carers. Evidence of a possible 'ceiling effect' in patients with dementia has been found in several other studies, with more than one-third of those diagnosed rating themselves at the highest level for several or all of the five EQ-5D dimensions.[35] Considering residents, one explanation for these results is that those with poor cognitive functioning or dementia have difficulty interpreting the questions and that in the presence of this difficulty, there is a systematic bias towards 'no problems'. If this is the case then this appears to be limited to the mobility, self-care and usual activities dimensions given a change in cognitive functioning and to the self-care and usual activities dimensions given a diagnosis of dementia in care home records. If it were found to be the case that such a systematic bias does exist for these participants for these dimensions, then the suitability of the EQ-5D as a measure in this population must be drawn into question. However, proxy reporting of the same measure does not solve the problem, as independently of MMSE score and a diagnosis of dementia in care home records, we have found proxy measurement to be insensitive to changes in resident health-related quality of life in several respects.

Previous studies have examined the inter-rater agreement between care home residents and carers using the OPERA data[2] and between multiple proxies in a population of patients with dementia.[36] Other studies of patients with dementia have explored the construct validity of data provided by different proxies, by exploring the association between clinical data and EQ-5D utility scores.[37] Furthermore, a recent review has summarised the evidence and issues surrounding the use of the EQ-5D in people with dementia.[35] The approach of this study is of a different nature, instead constructing statistical models for the EQ-5D, at both index (utility) score and dimension levels, and exploring predictors of discrepancies between residents and their carers by comparison of the models produced by the resident-reported and carer-reported EQ-5D data. These models show that the clinical characteristics of care home residents predict discrepancies between EQ-5D utility or dimension scores derived from health state descriptions provided by residents and their carers, in ways that are largely hidden when these discrepancies are aggregated across the population.

These findings question the ability of carers to make accurate assessments of care home residents' health-related quality of life for those with depression and/or pain, with consequences for any of the various parties who might have a stake in the reliability or accuracy of such assessments. These could include general practitioners, family members and policymakers wishing to assess the impact of care home interventions using health-related quality of life assessments made by carers. Furthermore, to the extent that carers are unable to

accurately assess health-related quality of life in certain subgroups, their ability to provide care that is responsive to patient needs may be compromised. As an example, depressed residents are more likely to report having problems with their mobility, self-care, usual activities and pain; however, for this same group of residents, carers are no more likely to report them having problems in these areas and are only more likely to report them as having problems in the anxiety or depression dimension. It may be reasonable to assume that if these problems are unreported by carers then they are therefore going unnoticed, making it less likely they will be addressed.

Care home staff appear to be most responsive to those elements of residents' health-related quality of life with which they are most involved, namely their activities of daily living. Problems relating to depression and pain, on the other hand, where the experience is largely internal, are more likely to go unnoticed, as is the scope of these effects in terms of the impact on other dimensions of health-related quality of life.

A potential limitation of this study is the fact that 504 participants in the OPERA study were excluded due to missing data, representing 48% of participants consented to OPERA at baseline (n=1060). The excluded participants differed from those with complete data in several respects, with statistically significantly 'worse' scores for proxy-reported EQ-5D utility, GDS-15, MMSE, SPPB, Barthel Index and SES, and with a greater proportion having a diagnosis of dementia in care home records. However, in a post hoc sensitivity analysis, following multiple imputation of missing data and recomputation of the analysis of the utility score level data, the results did not change in any important respect. In particular, for the regression on the differences between resident-reported and proxy-reported utility scores, the same variables were statistically significant, the signs were the same and the magnitudes of the statistically significant predictors were identical to two decimal places, with the exception of the coefficient for a diagnosis of dementia which was within 0.015 difference. The details of the multiple imputation procedure used and the model results computed using the imputed data sets are contained in online supplementary appendix A.

Responses to the EQ-5D-3L take the form of 'no problems', 'some or moderate problems' or 'serious or extreme problems' and so it is not known whether, when either 'some or moderate problems' or 'serious or extreme problems' is selected by residents but not by proxies, this is due to proxies not considering the problem to be sufficiently significant to warrant reporting or whether they are simply unaware that any problem exists. While the latter seems likely, further research, for example, using the EQ-5D-5L (with five response 'levels' as opposed to three), should help to confirm this. In order to further understand the mechanism by which discrepancies are generated, a qualitative study conducted alongside collection of the EQ-5D could aid in understanding the processes by which residents and proxies interpret 'problems' in each dimension and decide how to respond. The insight gained from such a study could lead to a clearer understanding of the likely impact of discrepancies on the care received by residents and inform further recommendations on how these discrepancies should be addressed.

Our findings are relevant to those making resource allocation decisions: data taken from residents or carers may produce different decisions, given the clinical profile of the population and the specific dimensions of health-related quality of life impacted by the intervention being evaluated. By identifying clinical predictors of this divergence, the results are also of use to researchers and decision-makers applying the EQ-5D in care home populations.

## CONCLUSION

Care home staff and residents differ fundamentally in their assessment of residents' health-related quality of life, as measured by the EQ-5D. This study has identified the factors most predictive of these discrepancies, and by examining how they are manifested at the level of the individual dimensions of health-related quality of life described by the EQ-5D, offers a deeper understanding of how they are revealed and when they are most problematic. The differences identified in this study may make it difficult for carers to act as the residents' advocate with health and social care professionals and could lead to interventions evaluated using proxy-based quality-adjusted life year estimates being wrongly rejected on cost-effectiveness grounds. A more resident-focussed approach to assessment of health-related quality of life is needed, as well as further research to better understand the potential differences in the mechanisms by which residents and proxies determine the relevant response to each EQ-5D dimension.

**Acknowledgements** The authors would like to thank the participants of the OPERA study, both the care home residents and the care staff, for providing the data for this study. The authors would also like to thank Dr Rachel Potter for her useful comments on the manuscript.

**Contributors** BP and JM designed the analyses for this study. BP conducted the analyses and wrote the manuscript. All authors contributed to the design of the study and to the review of the published material in this area, as well as providing critical input into the writing and revising of the manuscript. MU is the guarantor of the article.

**Funding** The OPERA trial on which this study was based was funded by the National Institute for Health Research Health Technology Assessment programme grant number 06/02/01. This study relates to the results of the OPERA trial, trial registration number ISRCTN43769277 (http://www.isrctn.com/ISRCTN43769277). The Warwick Clinical Trials Unit, University of Warwick, benefited from facilities funded through the Birmingham Science City Translational Medicine Clinical Research and Infrastructure Trials Platform, with support from Advantage West Midlands.

**Disclaimer** The views expressed are those of the authors and not necessarily those of the funding bodies.

**Competing interests** None declared.

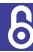

**Ethics approval** For the original OPERA study: ethical review for the study was provided by the Joint University College London/University College London Hospital Committees on the Ethics of Human Research (Committee A), now known as Central London REC 4. The REC reference for the study is 07/Q0505/56. The Committee also approved 10 Substantial Amendments for the study.

**Provenance and peer review** Not commissioned; externally peer reviewed.

**Data sharing statement** No additional data are available.

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
