## [Reviewer comments · BMJ Open]

ARTICLE DETAILS

TITLE (PROVISIONAL)	Can care staff accurately assess health-related quality of life of care home residents? A secondary analysis of data from the OPERA trial
AUTHORS	Parker, Ben; Petrou, Stavros; Underwood, Martin; Madan, Jason

VERSION 1 - REVIEW

REVIEWER	Judith Godin Nova Scotia Health Authority Canada
REVIEW RETURNED	20-Jun-2016

GENERAL COMMENTS	Review for "Can care staff accurately assess health-related quality of life of home care residents?" The discrepancy between proxy and care home resident quality of life scores is an important area of research for both clinical practice and future quality of life research. Much research has been conducted in this area and there is ample evidence that there are differences between proxy reports and resident reports of quality of life. The current paper contributes to this body of knowledge by examining predictors of the discordance between proxy-based and resident-based reports of quality of life. Overall the paper is well-written and the authors base their conclusions on thorough and rigorous statistical analyses. There are some issues I believe should be addressed before the paper is ready for publication and I detail these below. The authors open with a statement that the measurement of health-related quality of life of care home residents is problematic and the authors seem to end with the same conclusion. I think the authors could do more to explain their contribution to this area. That proxy and resident ratings of quality of life are not equivalent is not new knowledge. The conclusion of the paper should focus on the new contribution that this paper provides: a deeper understanding of when these discrepancies are particularly problematic and a better understanding of the factors that influence proxy versus resident ratings. I encourage the authors to discuss possible solutions for future research rather than simply stating that there is a problem. The authors reported that 40% of participants had a MMSE score that indicated severe cognitive impairment. They also discussed the possibility that residents with severe dementia may have difficulties understanding the questions and, thus, may respond "no problem" even when problems exist. This point merits more discussion. If this is the case should the resident reports be considered the 'gold standard' to which the caregiver's response is compared to? Further, it is in this population (severe dementia) that proxy reports are more
---

	likely to be used. “The dataset used for this study included 556 residents, comprising those individuals for whom both self-reported and proxy EQ-5D scores as well as data for the covariates used in the statistical models were available at baseline.” How many participants were excluded due to missing data? Were the participants excluded different in any way from those included in the analyses? The authors refer to categories of cognitive impairment (e.g., severe cognitive impairment) based on MMSE scores prior to providing the cut-off values. The cut-off values should be included when the categories are first mentioned. Although this is a secondary data analysis, the original study used human participants. I think it would be prudent to include the ethics information from the original OPERA study. More discussion is required regarding the limitations of the paper.
--	---

REVIEWER	Mary Ellen Dellefield VA San Diego Healthcare System, California, USA
REVIEW RETURNED	01-Aug-2016

GENERAL COMMENTS	This is a very interesting study that is well written and was a pleasure to read. My only comment is that assessment data were obtained from care staff, suggesting to an American reader that this refers to direct care workers or nursing assistants. It would be helpful to describe the training of care staff and their certification or licensing scope of practice. I mention this because in the USA, the RN is legally responsible for making assessments of residents, whereas the direct care workers are observers and data collectors working in partnership with the RN. One wonders if the care staff are suited to provide proxy reports in general. Also we do not know how familiar the care staff are with the resident. They may have just been working with the resident for one or two days. The accuracy of proxy-reporting presents challenges. It may be interesting to explore which staff are in the best position organizationally to provide this assessment. I am not suggesting that the RN is necessarily the most accurate reporter, but that the selection of the staff who are reporting is given greater consideration.
---

VERSION 1 – AUTHOR RESPONSE

Comment	Response*
Reviewer 1: The authors open with a statement that the measurement of health-related quality of life of care home residents is problematic and the authors seem to end with the same conclusion. I think the authors could do more to explain their contribution to this area. That proxy and resident ratings of quality of life are not equivalent is not new knowledge. The conclusion of the paper should focus on the new contribution that this paper provides: a deeper understanding of when these discrepancies are	We have amended the conclusion (page 17) to draw greater focus to the contributions referred to. In addition, we have added suggestions for further research in the discussion, highlighting the limitation that the differences in the mechanisms by which residents and proxies

particularly problematic and a better understanding of the factors that influence proxy versus resident ratings. I encourage the authors to discuss possible solutions for future research rather than simply stating that there is a problem.	determine their response to each EQ-5D dimension is unknown (page 16 paragraph 2), and suggesting a potential direction for future research to address this (page 16-17).
Reviewer 1: The authors reported that 40% of participants had a MMSE score that indicated severe cognitive impairment. They also discussed the possibility that residents with severe dementia may have difficulties understanding the questions and, thus, may respond “no problem” even when problems exist. This point merits more discussion. If this is the case should the resident reports be considered the ‘gold standard’ to which the caregiver’s response is compared to? Further, it is in this population (severe dementia) that proxy reports are more likely to be used.	We have added some discussion on this point (final two sentences of the first paragraph on page 14); however, we are reluctant to suggest the use of proxy-reported EQ-5D as a solution, for the reasons laid out in the rest of the paper.
Reviewer 1: “The dataset used for this study included 556 residents, comprising those individuals for whom both self-reported and proxy EQ-5D scores as well as data for the covariates used in the statistical models were available at baseline.” How many participants were excluded due to missing data? Were the participants excluded different in any way from those included in the analyses?	We now report in the first paragraph of page 6 that 504 participants in the OPERA study were excluded. The excluded participants showed statistically significant variation from our study participants with respect to some characteristics. We have added further details of this issue to the discussion (page 16) and carried out multiple imputation and repeated the utility score analysis on the imputed data, finding comparable results. These results have now been included as an appendix and used to justify the assertion that although the excluded participants were different in some respects, the dataset used remains broadly representative, given the similar results gained from the imputed dataset.
Reviewer 1: The authors refer to categories of cognitive impairment (e.g., severe cognitive impairment) based on MMSE scores prior to providing the cut-off values. The cut-off values should be included when the categories are first mentioned.	Table 1 is referred to in the second paragraph of the methods section and provides the cut-off values for the MMSE and the other outcome measures, and the only point before this that the categories are referred to is in the abstract.
Reviewer 1: Although this is a secondary data analysis, the original study used human participants. I think it would be prudent to include the ethics information from the original OPERA study.	This has been added to page 19.
Reviewer 1: More discussion is required regarding the limitations of the paper.	We have added points to the discussion

	covering the issue of missing data and of the unknown mechanism by which residents and proxies determine their response (pages 16-17).
Reviewer 2: This is a very interesting study that is well written and was a pleasure to read. My only comment is that assessment data were obtained from care staff, suggesting to an American reader that this refers to direct care workers or nursing assistants. It would be helpful to describe the training of care staff and their certification or licensing scope of practice. I mention this because in the USA, the RN is legally responsible for making assessments of residents, whereas the direct care workers are observers and data collectors working in partnership with the RN. One wonders if the care staff are suited to provide proxy reports in general. Also we do not know how familiar the care staff are with the resident. They may have just been working with the resident for one or two days. The accuracy of proxy-reporting presents challenges. It may be interesting to explore which staff are in the best position organizationally to provide this assessment. I am not suggesting that the RN is necessarily the most accurate reporter, but that the selection of the staff who are reporting is given greater consideration.	Thank you for your comments, unfortunately the information we have on the carers providing the assessments is limited to what has already been included in the paper, and we don't have any reliable data on how well the residents were known to the carer providing the assessment. The carer would have been the resident's key member of staff, or carer looking after them on the day of data collection. The point about which member of staff is best positioned to provide these reports is not one we are able to address in this study, given the limited data we have. Added two sentences (middle of paragraph 2 of methods, pages 5-6) to describe what is known about the carers
*all responses to comments have been highlighted in red in the main text	

VERSION 2 – REVIEW

REVIEWER	Judith Godin Dalhousie University Nova Scotia Health Authority Canada
REVIEW RETURNED	07-Nov-2016

GENERAL COMMENTS	The authors have addressed my concerns from the previous review: Missing data has been discussed and handled appropriately. Further discussion regarding residents with severe cognitive impairment has been included. Limitations of the study are discussed. The conclusion of the paper more clearly identifies the contribution of this work. Minor point for discussion: Consider the possibility that staff/proxy characteristics could also be associated with the discrepancies between resident and proxy ratings of health related quality of life.
---